# Biographical Reinvention: An Asset-Based Approach to Understanding the World of Men Living with HIV in Indonesia

**DOI:** 10.3390/ijerph20166616

**Published:** 2023-08-20

**Authors:** Nelsensius Klau Fauk, Lillian Mwanri, Hailay Abrha Gesesew, Paul Russell Ward

**Affiliations:** 1Centre for Public Health, Equity and Human Flourishing, Torrens University Australia, Adelaide, SA 5000, Australia; lillian.mwanri@torrens.edu.au (L.M.); hailay.gesesew@torrens.edu.au (H.A.G.); paul.ward@torrens.edu.au (P.R.W.); 2Institute of Resource Governance and Social Change, Kupang 85227, Indonesia; 3College of Health Sciences, Mekelle University, Mekelle P.O. Box 231, Tigray, Ethiopia

**Keywords:** biographical disruption, biographical reinvention, additive strategy, subtractive strategy, hope, optimism, resilience, men living with HIV, Indonesia

## Abstract

HIV diagnosis and management have often caused disruption to the everyday life and imagined futures of people living with HIV, both at individual and social levels. This disruption has been conceptualised, in a rather dystopian way, as ‘biographical disruption’. This paper explores whether or not biographical disruption of living with HIV encourages men living with HIV (MLHIV; *n* = 40) in Yogyakarta and Belu, Indonesia, to reinvent their sense of self and future over time using internal and external assets. Our analysis uses the concepts of additive and subtractive resilience strategies, and we show how, rather than having a purely disrupted biography, participants talked about their experiences of ‘biographical reinvention’. Study participants were recruited using the snowball sampling technique, beginning with two HIV clinics as the settings. Data were collected using one-on-one in-depth interviews, and a qualitative framework analysis was used to guide step-by-step data analysis. The findings showed that, despite the disruptions in their everyday lives (i.e., mental health condition, work, activities, social relationships, etc.) following the HIV diagnosis and management, MLHIV in our study managed to utilise their internal assets or traits (i.e., hope, optimism, resilience) and mobilised external resources (i.e., support from families, friends and healthcare professionals) to cope with the disruptions. An interweaving of these internal assets and external resources enabled them to take on new activities and roles (additive resilience strategies) and give up health compromising behaviours (subtractive resilience strategies). These were effective for most MLHIV in our study, not only to cope with the HIV repercussions and improve their physical and mental health conditions, but to think or work on a ‘reinvented’ biography which encompassed resilience, hope and optimism for better health, life and future. The findings indicate the need for HIV interventions and healthcare systems that provide appropriate support for the development and maintenance of internal assets of PLHIV to enable them to cope with the repercussions of HIV and work on a ‘reinvented’ biography.

## 1. Introduction

HIV diagnosis and management have often caused disruption to the everyday life and future of people living with HIV (PLHIV), both at the individual [1,2,3,4] and social levels [3,5], which Bury conceptualised as biographical disruption [6]. The concept of biographical disruption was initially introduced to describe a fundamental rupture in the fabric of everyday life of people living with chronic illness and how the illness experience disrupted their perception and understanding of themselves or life and the future [6,7,8]. The disruption could affect people’s habitual behaviours and patterns or routines that govern or structure their daily lives, which Bury called ‘the disruption of taken-for-granted assumptions and behaviours’ [6]. The disruption of taken-for-granted assumptions and behaviours will require people to profoundly rethink their concepts about their futures and contend with the disruptions of the illness to those futures, which he called ‘fundamental rethinking of a person’s biography and self-concept’ and mobilise external resources to respond to or to cope with disruptive impacts [6].

In this study, in addition to the stories of biographical disruption that HIV caused in their lives which resemble those previously reported [3,9,10], men living with HIV (MLHIV) also shared experiences of how they had been trying to cope with the disruption. In this paper, we use an ‘assets-based’ approach which tries to look for and understand positive assets MLHIV have; these items emerged in their interview narratives as well as how and why these have benefited MLHIV throughout their experiences of ‘reinventing’ their biography or future. Some studies have suggested that people facing difficult life events, including HIV infections and other health problems, undergo a psychological process which enables them to look for causal attributions, appraise the events and construct positive meanings for the events that could help them cope with the repercussions and rebuild their lives and futures [11,12]. Similarly, expectation for the best possible and positive outcomes for a better life and future is another positive internal aspect that can have a positive influence on people’s lives and enable them to avoid negative emotions or stress, depression and anxiety [13,14]. Previous findings also have reported self-restraints as enabling factors for people in difficult life situations to change their lives and give up activities that are incompatible with their new situations and which help them to reorganise their everyday lives and futures [6,15]. External resources also have been reported as enabling factors for PLHIV to cope with difficult circumstances or the mental health and social repercussions of HIV [6,16]. External resources may include supportive social networks and relationships with families, friends and neighbours that can help them cope with the disruptions and develop their futures [4,6,11].

It is evident in the participants’ narratives that the onset and management of HIV rupture the fabric of their everyday lives [6]. These were reflected in their experiences of poor physical and mental health conditions and the negative impacts on their activities, work and social relationships with families and others within the community. Despite such disruptions, internal assets and external resources are enabling assets for PLHIV throughout difficult experiences of living with HIV. Thus, this paper focuses on presenting these assets guided by the concepts of additive and subtractive resilience strategies [15]. It aims to show how, rather than having a purely disrupted biography, participants talked about their experiences of ‘biographical reinvention’ [15,17,18]. As the use of these strategies for ‘biographical reinvention’ in the context of HIV has never been reported, this is the novel contribution of our study to the knowledge about HIV management or coping. Thus, in this paper, we explored the men’s accounts of both the features of biographical disruption and how they integrated the condition of living with HIV into their lives by using additive and subtractive strategies to cope with the disruptions and ‘reinvent’ their biography or future.

## 2. Methods

### 2.1. The Concept of Biographical Reinvention

Building on the concept of ‘biographical disruption’, the concept of ‘biographical reinvention’ was introduced in research to understand different resilience strategies smokers used to quit smoking and remain non-smokers, or maintain their new identity of ‘non-smoker’ [15]. In their study, Ward et al. explain that, to quit smoking and remain non-smokers, people employed what they called additive and subtractive resilience strategies [15]. Additive strategies refer to people’s engagement in new activities, roles and practices in their individual lives or within the context of their community groups and organisations that could help them quit smoking or focus on activities other than smoking. Subtractive strategies are about giving up activities, practices and social relationships that reinforce smoking behaviours. The success of these strategies is also determined by the mobilisation or use of external resources (i.e., social support from families, friends, healthcare professionals, etc.) by individuals experiencing disruptions due to illnesses. Thus, additive and subtractive strategies often involve an interweaving of individuals’ internal assets or traits and external resources that enable people to take on health-promoting activities, practices and roles, and quit health-compromising activities, behaviours and social relationships [15,17,18]. The two concepts assisted us in analysing and understanding the ruptures HIV presents to the everyday life and social life of MLHIV and how they utilised their internal assets or traits and mobilised available external resources to develop better health, lives and futures. Thus, in this paper, biographical reinvention refers to the process through which the disruptions of living with HIV lead to shifts in the men’s identity through additive and subtractive strategies that are enabled by their internal assets and the available external resources. Guided by these concepts, we conceptualised how MLHIV think of or work on ‘reinvented’ biographies as people with resilience, hope and optimism for better health, lives and futures.

### 2.2. The HIV Epidemic in Indonesia and the Study Settings

Despite the declining trend of new HIV infections and AIDS-related deaths in Asia and globally [19], HIV infections in Indonesia have been reported to increase significantly during the last decade, from 55,848 cases in 2010 to 191,073 cases in 2015 and 526,841 in September 2022 [20]. The number of people whose HIV status progressed to AIDS also increased markedly, from 33,491 cases in 2010 to 83,241 in 2015 and 142,009 in September 2022; of these, 61,192 people have died from AIDS [20]. Of the total number of PLHIV in the country and those who are newly diagnosed every year, the majority are men [20]. For example, in 2022 alone (January–September), 71% of 36,665 new cases were diagnosed in men, which increased from 67% in 2020 and 70% in 2021 [20,21,22]. Risky sexual behaviours or sexual contacts have been reported as the main mode of transmission, and this is supported by the high prevalence of the infection (85.3%) being diagnosed in the sexually active age group (20–49 years) [20]. The increased HIV infections and AIDS cases in the country seem to reflect the low percentage of ART uptake among PLHIV. Of the total number of PLHIV in the country, 79% (417,778) knew their HIV status, and of these, only 41% (169,767) were on ART [20]. Of the ones who were on ART, only 29% (27,381) had their viral load suppressed [20]. The low prevalence of ART uptake seems to also reflect a low coverage of HIV care and treatment services, including HIV testing and ART, in Indonesia [20]. It has been reported that HIV care and treatment services are not available in several districts. Of the 504 districts in the country, only 476 have services and provide regular reports on service use [20].

Belu is located in Eastern Indonesia and shares its border with East Timor [23]. It has a total population of 204,541 people [23]. It has 17 public health centres or sub-public health centres, three private hospitals and one public hospital; the only HIV clinic in the district is located there [23]. HIV care services available at the clinic are limited to HIV counselling, testing and antiretroviral therapy (ART). Liver and kidney function tests, CD4 tests and viral load tests to support ART or measure the effectiveness of ART are not available. In terms of HIV infections, it is reported that there had been 1200 cases diagnosed in the district, and of these, only about 25% ever started ART at the clinic [23]. The limited availability of HIV-related healthcare services is one of the barriers to HIV testing and ART uptake among the general population and PLHIV in the district [3]. Yogyakarta municipality is part of the Special Region of Yogyakarta province, and has a total population of 636,660 people [24]. It has a total of 1488 PLHIV, and of these, the majority were on ART [25]. The healthcare facilities available in the district include 20 hospitals and 27 public health centres and sub-public health centres [24]. HIV care services, such as HIV counselling and testing (VCT), CD4 and viral load tests, ART, and other medical tests to support HIV treatment, are provided in 10 public health centres and four hospitals [26].

### 2.3. Participant Recruitment and Data Collection

Data reported in this paper are part of a large-scale qualitative study that sought to understand HIV risk factors, impacts and services use among PLHIV in Yogyakarta and Belu, Indonesia. Using the snowball sampling technique, initial participants were recruited through HIV clinics in each setting after securing permission letters from the clinics. Data collection began with the distribution of information sheets containing details about the study and the field researcher’s contact numbers through the receptionists and information boards of the clinics. Participants stated their intention to participate through short message service (SMS) or phone calls. An initial conversation was made between the field researcher (NKF) and each potential participant to discuss and agree upon the interview time and venue. The initial participants who contacted the researchers to confirm their participation were recruited and interviewed. After the interviews, each of them was asked to recommend or distribute the study information sheets to other eligible people, such as their families, colleagues and friends, who were interested and willing to participate in the study voluntarily [27]. The recruitment process took six months (June–November 2019), with 40 MLHIV finally participating in the study. The sociodemographic profile of the participants is provided in Table 1.

One-on-one in-depth interviews with participants were conducted in a private room at the HIV clinic in Belu and a rented house nearby the HIV clinic in Yogyakarta. The interviews were between approximately 35 and 87 min long, performed in Indonesian (the primary language of the participants and the researcher, who also speaks fluent English) and audio recorded using a digital recorder. For this particular topic, the interviews explored participants’ perceptions and experiences about disruptions HIV presented in their daily individual and social lives and how they utilised their internal traits or capital and mobilised external resources to cope with those disruptions and develop better health, life and future. Data saturation and the richness of information were used to justify our decision to cease participant recruitment and interviews. These were reflected in the similarities of responses by the last few participants to those of previous ones. One person in Belu withdrew his participation for personal reasons after the interview ran for 15 min and the recording was deleted and excluded from this analysis. No repeated interview was conducted with any of the participants. Due to the sensitivity of the information collected during the interviews, we decided not to offer opportunities for participants to read, correct and provide feedback on the interview transcripts. This was to avoid the possibility of divulging HIV status of the participants who may not have disclosed it to their family members.

### 2.4. Data Analysis

Before the comprehensive analysis, the audio recordings were transcribed verbatim by the first author (NKF). Transcription was initiated alongside the data collection process, and notes taken during the interviews were integrated into each transcript during the transcription process. The analysis was guided by a framework analysis for qualitative data by Ritchie and Spencer, which suggests several steps of qualitative data analysis [28]. Data analysis was performed in Indonesian, which helped to keep the sociocultural meanings attached to the information provided by the participants [29]. (i) The transcription process and reading the transcripts repeatedly during the analysis allowed the researcher to become familiar with the data, provide comments on data extracts, and break down the information into small chunks; (ii) During the process, key concepts and issues identified from the transcripts were listed and then used to form a thematic framework. The identification of the thematic framework was an iterative process that involved changing and refining themes; (iii) Each transcript was indexed by providing open codes to data extracts, followed by close coding to identify and group similar or redundant codes into the same themes and sub-themes, which were informed by the additive and subtractive strategies. Comparison of the findings (codes and themes) within and across interviews was repeatedly performed throughout data analysis; (iv) Finally, the entire data were mapped and interpreted as presented in this manuscript. The selected quotes for this publication were translated into English by NKF and then checked by other authors for clarity. The process of checking and rechecking quotes against the translated interpretations or examination of meaning in both languages was also performed to maintain the accuracy of the translation and credibility of the findings [30]. Authors regularly undertook discussions, comments, feedback and revisions during the analysis and writing process and agreed on the final themes and interpretations presented in this paper.

The ethics approvals for this study were obtained from the Social and Behavioural Research Ethics Committee, Flinders University (No. 8286) and the Health Research Ethics Committee, Duta Wacana Christian University (No. 1005/C.16/FK/2019). For de-identification purposes, all personal information was removed from each transcript. Each transcript was given a letter and number, such as PY1, PY2, … (PY = participant from Yogyakarta) and PB1, PB2, … (PB = participants from Belu).

## 3. Results

### 3.1. Sociodemographic Profile of Men Living with HIV

The mean age of the participants was 38.12 years, with the majority in the age group of 30 to 49 years (*n* = 30) (see Table 1). Just over half of them had been diagnosed with HIV from two months to 5 years ago (*n* = 21), while the rest had been living with HIV for a longer time, between 6 and 10 years or more. The majority were (re)married (*n* = 25), while the rest were unmarried/single or divorced or widowed. The majority graduated from either junior or senior high school, some graduated from elementary school and the rest had graduate diploma certificates. Most of them had either paid jobs or their own businesses, while the rest had unpaid jobs (i.e., farmers) or were unemployed or had quit jobs due to physical and health-related issues following the HIV diagnosis.

The findings were grouped into three main themes centred around hope, optimism and resilience: (i) HIV consequences on the participants’ lives; (ii) hope for a better health condition and future; (iii) optimism about recovery and work; and (iv) resilience towards HIV infection. The details about themes are presented below.

### 3.2. HIV Consequences on the Participants’ Lives

HIV diagnosis and management, especially at the early stage of the infection, disrupted the everyday life of most participants, including disrupting their physical and mental health conditions, activities and social relationships with others within families and the communities where they lived. These are coherent aspects that weave the fabric of their everyday lives. Thus, the disruption to any or all of these aspects would negatively impact their lives. For example, the following narratives reflect on how the early stage of HIV diagnosis negatively affected the participants’ physical and mental health conditions, which seemed to also be influenced by their incognisance of the infection, HIV care services and the available social support they could access:

“*After the diagnosis, my life turned upside down 100%. You can see my body gets skinnier, I’m physically weaker day by day. I feel stressed, anxious, and angry for no apparent reason. I felt like my life fell apart … I thought my life was over. I would die soon. I did not know about peer support groups and the treatment process. The nurse talked to me about ART but I did not pay attention because I was diagnosed with HIV when I was very sick and admitted to this hospital …*”(PB5)

“*I was very weak physically. My CD4 count was one. My body was just the skin and skeleton and was very black all over. I was so scared. It was a very stressful moment. My life was dark. It was so difficult and I did not know what to do, who could help me … I felt like HIV defeated me. My ambition to be this and that was over, something like I was a loser …*”(PY17)

For some participants, having poor physical and mental health conditions influenced how they perceived themselves, which was in a more negative way: ‘*I was a burden to my parents* [his wife’s family sent him back to his parents after he was diagnosed with HIV]’ (PB1), ‘*I felt like I was a useless person …*’ (PY13). Participants also talked about the disruption of their ability to work and social relationships with family members and friends following HIV diagnosis: ‘*I had to stop working (quit) because I was very weak physically*’ (PY8), ‘*I knew that people (who knew about his HIV status) stay away from me*’ (PB20). For some, such conditions seemed to also contribute to the negative self-perceptions about themselves as people who could not provide for themselves and were unwanted by others within families and others in their social relationships:

“*I felt like others did not want me in my family. My parents hardly talked to me. They did not call me out of my room when they were eating. I’m a grown-up person but I felt like I was neglected during those difficult times … Some of my friends stayed away from me. When they gathered and saw me coming then they would leave one by one*”(PB18)

“*I experienced it (discrimination), it felt painful here (pointing to his chest). I was ostracised by family: my father, mother, brothers and sisters. At that time, I did not have the spirit to live anymore because I was ostracised like that. It took place for a long period of time till I remarried. They did not want me in the house. At the first time I told them that I contracted HIV, everybody was shocked and angry at me. I was nearly chased away from home. … I was scolded and asked not to use the toilet, if I used it then I had to clean it up afterwards (the man and his second wife lived apart from his family after they got married)*”(PY19)

### 3.3. Hope for a Better Health Condition and Future for Themselves and Their Families

Despite the disruptive experience of these aspects, as frequently emerged in the narratives of our participants, their ability to make use of internal assets or traits, such as hope (i.e., hope for better health and future), was part of the process of coping with the disruptive experience and ‘reinvent’ their biography or imagined future. For example, a strong hope for better physical and mental conditions was often mentioned by several participants across the study settings, especially those with poor physical and mental health conditions when the study was conducted. ‘*Better health condition*’ was narrated as a goal, and a strong hope guided their focus and attention towards the goal, as one of them explained: ‘*I am fully focused on recovery; that is my goal*’ (PB9). A better health condition, by these participants, was perceived as a new starting point of the process of beginning a better life or reinventing the self: ‘*I want to be healthy, and that would be a new phase to start for a better future*’ (PY11).

Such a strong hope also guided their plan and commitment to achieving their goal (better health condition), and this can be seen in the compliance with the treatment or antiretroviral therapy (ART) they have started before the study. Some expressed: ‘*I adhere to the treatment because I want to be healthy*’ (PY6) and ‘*what the doctor says, I will listen and do it*’ (PB3). External resources, such as information support from healthcare professionals, was also employed and played a significant role in the participants’ biographical reinvention process. The use of information about the effectiveness of ART that was shared by health professionals to amplify commitment to the treatment was an instance of the mobilisation of external resources for self-support and enable them to think of or work on ‘reinvented’ biography or future. Such information contributed to increased awareness and knowledge about ART as the only way to regain and maintain good health and helped reduce the emotional repercussion of HIV diagnosis:

“*I have a strong hope to be healthy again like before because it’s been five months since I’ve been very committed to treatment [ART] and have never been negligent like before. I collect the medicines every month and take my medicines every day. The doctor said this is the only way to regain my physical health and strength. So, I try not to get worried or stressed out. The doctor said there is no need to stress because the medical treatment could help me recover. This helps me calm down and not get stressed too much. All I want is to continue adhering to the treatment because I want to get better*”(PY12)

“*The virus is no longer detected in my blood [viral load suppressed], so I must maintain my health. Treatment is number one. I don’t want to miss taking the medicines. I always take my medicines on time every day. I have to be committed to the treatment. It is good because when I’m healthy like this, I don’t get stressed or worried …*”(PB17)

The focus on and compliance with HIV treatment or ART indicated participants taking on new routines (i.e., regular ART access, daily medication intake and regular check-up) and integrating them into their everyday lives. The engagement in such new routines or activities is characteristic of an additive strategy useful for the biographical reinvention, and it supported them in coping with the physical and mental health and social repercussions of HIV that they faced. Engagement in medication-related routines was also supportive in maintaining the awareness of a healthy lifestyle and helped them withdraw from or avoid risky behaviours that contributed to ill-health or HIV infection progression. The latter reflected the subtractive strategy which was also crucial in supporting the participants’ biographical reinvention or their efforts to develop a better future. The use of these strategies was reflected in the following narrative of a participant who talked about being fully committed to ART and was aware of healthy lifestyles, and who gave up health-compromising behaviours:

“*I adhere to the treatment [ART], never missed the medicine even for one day. Medication is a part of my routine. I have been doing the treatment for about two years, so I’m fully aware of the time to take medicine, rest, and sleep. It makes me always aware of my health condition and what to do and not to do to stay healthy. I don’t smoke, drink [alcohol], or take drugs like I used to do before the diagnosis and treatment. The doctors and nurses always remind me of these things, so they are always in my mind. It is good because when I feel physically healthy like this, I don’t get stressed, and my work is not disturbed*”(PY20)

Apart from the physical and mental health-oriented hope, motivation to have a better life and future for themselves and their families was another goal to achieve, which played a role in guiding the attitudes and behaviours of the participants towards the infection and their futures. This was another instance of subtractive strategy as an individual was able to leave behind the ruptures HIV presented to their lives and look forward to rebuilding their futures or identity: ‘*I always try to put away all the negative thoughts, feelings and experiences I have been facing and focus on the future. I’m doing good so far*’ *(PY7)*. It also reflected a sense of responsibility for themselves and family and a focus on a more family-oriented goal which was beyond themselves. Some said: ‘*I’m hoping for a better life for my family’* (PB19) and ‘*the future of my family is my responsibility*’ (PY1). Such hope or motivation was an important psychological booster for the management of HIV-related psychological repercussions, such as fear, stress, worry and depression. It is an internal asset or trait that redirected one’s focus from the negative impacts of HIV to the treatment and a better life and future for themselves and their families. The narratives of most married men who were also fathers across the study settings reflect such motivations or hopeful thoughts, which seemed to be supported by the awareness of their responsibility as the breadwinner for their families:

“*After the doctor told me that I’m infected with HIV, I was shocked, stressed, scared, and worried. I was so sad, and all kinds of negative thoughts were always in my head. I felt broken for months, but I am grateful that with the advice and support from some of my friends and HIV counsellors, I could get up. I thought I must keep working, do my activities, and continue supporting my life and family. I hope to be healthy again and strive for a better life for myself and my family. My family needs me. I am responsible not only for myself but also for my family. I am the family’s breadwinner; if I don’t fight, then what will happen to my family*”(PB10)

“*I hope to be able to prepare for my children’s future. So, I am now trying to continue supporting their education until they finish college. This is my focus. I experienced tremendous pressure when I was diagnosed. I felt ashamed, stressed, and afraid, but after a while, I realised my children need me to prepare for their futures. So, I try to overcome all the pressures and focus on my kids’ life and education. I work passionately for the sake of the children, not for myself*”(PY16)

### 3.4. Optimism about Recovery and Work

Optimism was another positive internal asset or trait underpinning the expectation for the best possible and positive outcomes for the majority of men we interviewed in this study. As mentioned in their narratives, optimism about the high possibility of achieving health recovery was amplified by the initial positive effects of ART on improving their physical health. For example, some participants explained: ‘*The medicines [ART] are very helpful*’ (PY5) and ‘*My body is getting stronger because of the treatment*’ (PB11). Thus, such optimism was also a reflection of a positive evaluation of the effectiveness of the treatment and the firm belief in the high possibility of physical and mental health recovery. It also showed the participants’ ability to positively attribute the initial successful results of the treatment with the possible future success if adherence to the treatment is maintained. Moreover, such optimism seemed to be a positive indication of the mental strength to overcome or manage the psychological and social repercussions of HIV infection, which could be an early sign of a successful biographical reinvention:

“*When I started [ART] treatment a year and a half ago, my body was covered in itchy sores, and I was scratching. However, after several months of receiving this medical treatment, I no longer feel itchy, the itchy sores have healed, and I feel physically strong. I feel positive, and this motivates me to continue antiretroviral therapy so that my viral load is suppressed and I don’t have to be worried about passing the virus on to my wife*”(PY9)

“*I’m optimistic that I will be fine and my health condition will recover. I was diagnosed with HIV last year [2018]. I started antiretroviral therapy seven months ago, I was very weak physically, but now you can see I’m getting stronger physically. I do not want to get stressed anymore; I want to enjoy my life and continue the therapy because it works for me, and I am optimistic that I will get better*”(PB14)

The latter participant from Belu also linked his optimism for regaining physical health or strength with new activities he had just engaged in before the interview, including being part of a choir group that had regular singing practice and taking on a role as a committee member of football competition at the sub-district level. As this activity and role required physical strength and mobilisation from one place to another for coordination, his ability to engage in such an activity and role was a sign of physical health improvement compared to his condition before ART initiation. Thus, this underpinned his optimism and was seen as a positive trajectory for recovery. Taking on the new activity and role was an additive strategy that supported his biographical reinvention process and helped him to regain his self-belief, as portrayed in the following narrative:

“*When I was diagnosed with HIV, I was terrified and I thought I would not be to do anything else. My body was frail and, in my mind, if someone gets HIV, he will be sick forever and just waiting to die. He will be more in bed and unable to work or move properly. Being involved in regular choir and singing practice and taking part as a committee member for the ongoing football matches, I have become more confident that I can still do whatever activities I want, and I am even more optimistic that I will recover*”(PB14)

Participants who had a job also showed strong optimism about keeping their jobs. Some optimistically expressed: ‘*I’m positive about keeping my job, as long as I want, I can keep my job for years ahead, and I’m doing great with my work*’ (PY13). The participants also talked about the support they received from co-workers and families, which were external resources that helped them be optimistic and continue with their jobs. At the same time, the interweaving of optimism and external resources (support from co-workers and families) helped them cope with adversities, especially psychological repercussions, such as stress, worry and concern:

“*I’m working for an NGO providing support for people living with HIV. Some of us [staff] are living with HIV, and others are not. We [the man and his co-workers] know and support each other. I’m happy with my work*”(PY5)

“*My parents and sisters, especially my oldest sister, who is a nurse, are very supportive. They support me to work so I don’t get stressed by being at home and doing nothing. They take care of me, monitor my medication, remind me to collect the medicines. That’s why I’m now healthy and continue to work and earn money for myself*”(PB4)

### 3.5. Resilience: Accepting, Finding the Meaning of Challenging Life Experiences and Bouncing Back

Resilience in coping with and adapting to the complex and challenging life circumstances related to HIV, primarily through mental, emotional, and behavioural flexibility to ‘bounce back’ from adversities, was part of the biographical reinvention of participants in both study settings. The findings highlighted the resilience in participants’ ability to face and accept disruptions or negative experiences of HIV and to find positive meanings in those experiences. For example, several participants across both settings described how they endlessly try to adapt their daily lives and activities to the condition of living with HIV, reflect on the related challenges and find ways to manage them. Although these were described as ‘*a very challenging task*’ (PB2), ‘*hard work to accomplish*’ (PY19) and ‘*never ending effort*’ (PY2), these seemed to build strong courage among the participants to cope with HIV-related negative consequences. The following narratives of two men who had been living with HIV for 3 and 5 years reflect how they managed to overcome the repercussions HIV presented to their lives:

“*I felt tremendous pressure in the first year after being diagnosed. But I have been trying to accept the situation [HIV positive status] and learn to live with HIV as normal. It’s not easy; I can feel it now. I try to think positively and strengthen my mind and heart to live with HIV in my body. Gradually the fear, anxiety and stress decrease until now*”(PY3)

“*It was a tough fight I have gone through to be able to accept my [HIV] status and to feel like a normal [non-infected] person. I guess I succeeded in passing through those difficult moments, and here I am. I do not overthink this infection anymore. I am living a normal life like others*”(PB1)

Efforts to explore the positive meanings of HIV-related disruptive experiences were also part of the resilience towards the infection. The efforts—to interpret such negative experiences in a positive way—were seen in their attitudes and actions to understand what the challenges meant to their lives and families. Such efforts led to different pathways of thoughts and reflections and a conclusion to change their attitudes and behaviours (i.e., change negative patterns of sexual behaviour), become helpful to others, such as families and friends, and look at the future positively. These illustrated both subtractive resilience strategies through giving up negative behaviours and additive resilience strategies through engagement in new activities to help families and others:

“*I got HIV because I had sex with many casual partners every place I went. I think the infection is a warning for me to change the bad pattern of my sexual behaviour and be faithful to my wife. I used to spend money to have sex with sex workers, but after I got HIV, I prioritise my family’s needs. I am also involved in volunteer activities to help other friends [PLHIV] and support them to access treatment, so I feel useful to them. Doing this makes me feel less stressed or burdened*”(PY10)

The narratives of several participants in Yogyakarta and Belu also showed how they were able to overcome various HIV-related challenges and successfully bounced back from the difficult circumstances they faced following the diagnosis and got stronger mentally. Their stories informed us of how they rose from difficult situations during the early stage of HIV infection and tried to return to the pre-HIV state. This was reflected in the new perception of themselves as much more organised persons and how they cared more about their health and work compared to pre-HIV diagnosis:

“*I want to show people that being infected with HIV is not an obstacle to being successful in business. For a year [during the early stage of the diagnosis], I experienced psychological shocks; stress, depression, and worry. Many negative thoughts came to my mind. But I tried to cope with all these challenges and improve myself and my work. Now my business is growing, and my income is increasing*”(PY19)

“*Contracting HIV is not a good thing, and I don’t want to get it, but I am grateful because my life has become more organised after the diagnosis. I care about myself and my health. I know when to take medicine, rest and work. I feel like my life is more balanced. Before being diagnosed with HIV, I didn’t care about my health condition. I got drunk often, smoked cigarettes, and had sex with sex workers without thinking of the consequences, but now I am always aware of what I do …*”(PB5)

The latter participant expanded his perception of being a more organised and health-caring person by talking about his self-consciousness about what he did in his everyday life. He mentioned about ‘*think before doing*’ which reflects his self-consciousness of activities or routines he engaged in on a daily basis. This was an indication of being able to overcome HIV-related disruptions and aware of when an activity should be undertaken for the sake of health or in his term, ‘*there is time to work, eat, play and rest*’, which is also an indication of being a self-organised person.

## 4. Discussion

This paper presents the use of additive and subtractive resilience strategies as enablers for biographical reinvention which have not been reported in the context of HIV [15]. Our participants’ experiences of biographical disruptions (i.e., poor physical and health condition, inability to work, problems with social relationships, etc.) brought about by HIV onset and management are not dissimilar to those of other PLHIV in different settings globally, and such experiences have led to their negative perceptions about themselves (i.e., loser, useless, neglected, unneeded person, etc.) and their lives (i.e., (stressful life, dark life, painful life, etc.) [16,31,32]. However, our study highlights novel findings on the application of additive and subtractive strategies that involves an interweaving of individuals’ internal assets or traits, such as hope, optimism and resilience, and external resources, including support from families, healthcare professionals, friends and co-workers, in the participants’ biographical reinvention. It highlights how these strategies enabled these men to work on a ‘reinvented’ biography as people with internal assets, such as resilience, hope and optimism, that enabling factors to achieve better health, life and future for themselves and their families.

Hope represents the participants’ motivational state that supports their desires for goals (i.e., better health condition and future for themselves and their families) and directs them towards those goals [33,34]. Thus, the recognition of these goals to be achieved despite the disruptions HIV presented to their lives was the underlying reason for the strong hope among the participants with poor physical and mental health conditions when the interviews were conducted. Such hope (for a better health condition) enabled them (i.e., to think of or work on a ‘reinvented’ future) for better health is perceived as the prerequisite or starting point for a ‘reinvented’ future. This reflects, despite various challenges facing them, a more positive way of perceiving the condition of living with HIV, which certainly guides their positive attitudes and behaviours towards positive outcomes through the available support and treatment. Similarly, hope for a better health condition enabled the participants, especially the married ones, who were aware of their responsibilities for their families and the future of their children, which could only be possibly fulfilled if they have a better health condition. This sheds light on how family situation could be a strong supporting factor or source of motivation for recovery for PLHIV, which is different from many other previous reports on rejection, refusal, ostracism and stigma against PLHIV within families [5,35,36]. As reflected in our findings, the men’s hope for a better health condition and future is intertwined with or underpinned by the mobilisation of external resources, such as informational support from healthcare professionals. These findings provide an understanding of how an interweaving of these aspects (internal asset—hope and external resource—health provider support) enabled participants to take on new positive activities (i.e., regular ART access, daily medication intake and regular check-up (additive strategy)) and giving up health-compromising activities (i.e., cigarette smoking, alcohol drinking, and drug use (subtractive strategy)). The application of these strategies enabled them to cope with or leave behind negative experiences (mental health and social repercussions of HIV) and work on a ‘reinvented’ future [15,32,37]. The findings have significant implications for HIV interventions and healthcare providers to recognise internal assets that PLHIV have and provide appropriate support to amplify those assets and enable PLHIV to cope with HIV-related challenges and develop their futures. In addition, our findings indicate the importance of PLHIV’s self-recognition of their internal assets, which can strengthen them psychologically to take on health-promoting behaviours, activities and roles to maintain their physical and mental health and well-being and cope with difficult circumstances [15,38].

This study has also highlighted the link between optimism [13,39] and successful biographical reinvention, which is another novel finding of the current study. Optimism for recovery, for example, arose from the participants’ positive evaluation of the effectiveness of ART as reflected in the improvement of physical health condition, and this enabled them to see a ‘reinvented’ future through the therapy. Similarly, optimism for a positive future job, stemming from the recognition of their positive job performances, enabled them to see a positive ‘reinvented’ future or prevented them from worrying about the future. Thus, the study also highlighted the importance of attributing the present positive condition as resulting from the initial ART and connecting that to future positive outcomes of the therapy while recognising other factors that support positive emotions and encourage actions to overcome the mental health repercussions of HIV [13,14].

However, it should be noted that the ability to make attribution and identify helpful strategies and actions are also determined by other factors, such as education, health literacy and social support from others [40,41]. Findings from previous studies with different population groups of PLHIV have suggested that PLHIV with good HIV health literacy and who receive or have external resources, such as emotional and social support from families, friends and healthcare professionals, have better access and adherence to ART and better psychological or mental health and well-being [3,42]. On the contrary, other studies have reported that health illiteracy (i.e., a lack of knowledge of ART and its effectiveness) and the unavailability of HIV services and social support are both barriers to HIV treatment and contributing factors for depression, stress and anxiety among PLHIV [43,44,45]. Our findings also report the undertaking of additive strategy through the participants’ engagement in new activities and roles and an interweaving of optimism and external support from co-workers and families that strengthened their self-belief in recovery and enabled them to cope with adversities and develop their futures. It is therefore crucial that HIV information or education for PLHIV, which covers the linkage to treatment and ART adherence and its benefits, needs to be strongly emphasised in HIV interventions, services and the healthcare system. This may result in positive behaviour changes towards HIV treatment or ART, as previously reported [46]. The findings also inform HIV program implementers of the importance of establishing social support networks of families, friends, healthcare professionals and co-workers for PLHIV, which could provide them with external resources to help them cope with psychological and social repercussions of HIV and work on a ‘reinvented’ future [47,48].

Our findings also highlight strong resilience in the men’s ability to accept their HIV-positive status and find positive meanings in a challenging situation [14]. This is consistent with the findings of previous studies with other population groups of PLHIV [49,50,51], reporting self-acceptance and self-reliance as supporting factors in coping with HIV-related mental health repercussions. The results reveal self-acceptance and finding meanings as enabling factors that led to the change in negative attitudes and behaviours (i.e., sex with sex workers) and the commitment to help other PLHIV (i.e., assisting others in accessing ART). These reflect the core aspects of both subtractive and additive resilience strategies and the participants’ self-realisation as external resources for others in need [15,32]. Thus, it is plausible to suggest that accepting HIV-positive status and meaning-making may assist men in overcoming challenges and improving their mental health conditions, which may also have positive impacts on the overall health and well-being of their family members.

Finally, it should be noted that, as is the case for many qualitative studies, the qualitative design used in this study was to help gain rich and in-depth personal narratives (i.e., information, knowledge, understanding, interpretation) of the participants on the topics being researched. It was not designed to claim the generalisability of the findings to all participants or other PLHIV. However, the richness and in-depth information presented in this study can be useful for the development of policy and evidence-based interventions to support PLHIV to rebuild a positive life following their experiences of various HIV-related negative impacts.

### Limitations and Strengths of the Study

There are some potential limitations that should encourage caution when interpreting the findings of this study. The starting points of recruitment through HIV clinics and the use of the snowball sampling technique might have led us to recruit participants from the same networks and who were on ART. It is, therefore, possible that PLHIV who were not on ART and were outside of the current participants’ social networks could have been under-sampled. Thus, their HIV-related perceptions and experiences, which might differ from those of the current participants, may not have been covered in the present study. In addition, as the focus of the study was on heterosexual men, we did not include high-risk or key population groups, such as men who have sex with men, male sex workers and male drug users, who have been reported to suffer double burden due to the infection and their sexual orientation and activities. However, we acknowledge that while some unknown number of men in the sample likely are members of a HIV key population (or engage in behaviour consistent with such membership), this possibility was not considered in the analysis of biographical reinvention and is worth exploring in future research. They may face different experiences and challenges and have different methods of managing the consequences of HIV and rebuilding their lives and futures, which are worth exploring in future research. However, the study’s strength is that this is an initial qualitative study that explores the in-depth biographical reinvention of MLHIV through additive and subtractive strategies. The coherent and structured way of qualitative data management to enhance transparency, rigour and validity of the analytic process as guided by the framework analysis is another strength of the study. Further large-scale studies to explore the internal assets of PLHIV and how those assets help them cope with HIV challenges and work on their ‘reinvented’ biography or future are recommended. Also, as the participants in the current study were on ART, further studies on this topic need to involve PLHIV who are not on ART and compare their findings to those presented in this paper.

## 5. Conclusions

This paper uses an ‘assets-based’ approach which attempts to find positive assets that MLHIV have. It presents how MLHIV coped with the biographical disruption HIV presented to their lives and utilised internal assets, such as hope, optimism and resilience, and external resources, including support from families, friends, co-workers and healthcare professionals, to enable them to take on new activities and roles (additive strategies) and give up health-compromising behaviours and practices (subtractive strategies). These were effective for most MLHIV in our study, not only to cope with the HIV repercussions and improve their physical and mental health conditions, but also to ‘reinvent’ their biography or future. The findings indicate the need for HIV interventions and healthcare systems that provide appropriate support for the development and maintenance of internal assets of PLHIV, that can enable them to cope with the repercussions of HIV and think of or work on a ‘reinvented’ biography or future. Interventions to build social support networks for and/or among PLHIV are also recommended.

## Figures and Tables

**Table 1 ijerph-20-06616-t001:** Sociodemographic profile of men living with HIV.

Characteristics	Men Living with HIV	
	Yogyakarta (*n* = 20)	Belu (*n* = 20)
**Age**		
20–29		7
30–39	10	5
40–49	10	5
50–59		2
60–69		1
**Marital status**		
Single	5	7
Divorced	2	
Widowed/r		1
(Re)Married	13	12
**HIV diagnosis**		
˂1–5 years ago	6	15
6–10 years ago	7	4
11–15 years ago	7	1
**Religion**		
Islam	17	
Catholic	3	19
Protestant		1
**Education**		
University graduate/Diploma	7	2
Senior high school graduate	11	8
Junior high school graduate	2	4
Elementary school graduate		6
**Occupation**		
Entrepreneur	10	1
Teacher		2
Farmer		3
Police		1
Private employee	5	1
Retired civil servant		1
University student		1
Taxi/Motor taxi driver	1	2
Iron welder		2
Mechanic	1	
Unemployed/quitting jobs	3	6

## Data Availability

The data presented in this study are available on request from the corresponding author. The data are not publicly available due to restrictions set by the human research ethics committee.

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
