# Peer review of "Biographical Reinvention: An Asset-Based Approach to Understanding the World of Men Living with HIV in Indonesia"

_ijerph, 2023, doi:10.3390/ijerph20166616_

Round 1

Reviewer 1 Report

Looking at people living with HIV from the perspective of "biological reinvention", which has been used to conceptualize adjusting to other chronic diseases, is a novel way to describe adjusting to a HIV diagnosis. The introduction to "biological reinvention", from lines 40-119, is repetitive and should be shortened. In its place, it would be helpful if the authors included a paragraph describing the HIV epidemic in Indonesia--epidemiology, the availability or lack of availability of HIV testing and treatment, etc. This would provide a context for the qualitative interviews. To give an example, if there are low rates of HIV testing and treatment in the areas of Indonesia where these interviews took place, then the authors may have interviewed subjects from only that subgroup of people who are more successful at getting into care than most people living with HIV. A related point is that none of the interviews touch on subgroups of people in the most high-risk categories, such as men who have sex with men or injection drug users. This should be described in the limitations section of the paper.

The paper is reasonably well written but needs further editing. For example, line 49 uses the words "contend the disruption" which should read "contend with the disruption". Line 53 uses the words "presented to their life" and should say "presented to their lives". These errors, especially missing prepositions, and the use of plural pronouns with singular nouns, occur through out the paper. The authors have done a great deal of work without funding for their research, but maybe they can find someone to correct the the English grammar and usage.

Author Response

Reviewer 1

Comment

Looking at people living with HIV from the perspective of "biological reinvention", which has been used to conceptualize adjusting to other chronic diseases, is a novel way to describe adjusting to a HIV diagnosis. The introduction to "biological reinvention", from lines 40-119, is repetitive and should be shortened. In its place, it would be helpful if the authors included a paragraph describing the HIV epidemic in Indonesia--epidemiology, the availability or lack of availability of HIV testing and treatment, etc. This would provide a context for the qualitative interviews. To give an example, if there are low rates of HIV testing and treatment in the areas of Indonesia where these interviews took place, then the authors may have interviewed subjects from only that subgroup of people who are more successful at getting into care than most people living with HIV. A related point is that none of the interviews touch on subgroups of people in the most high-risk categories, such as men who have sex with men or injection drug users. This should be described in the limitations section of the paper.

Response:

  • We have revised the introduction to minimise the repetition of the concept of ‘biographical reinvention’ (the detailed description of the concept is presented in the methods section).
  • The HIV epidemic in Indonesia has been provided

Despite the declining trend of new HIV infections and AIDS-related deaths in Asia and globally [26], HIV infections in Indonesia have been reported to increase significantly during the last decade, from 55,848 cases in 2010 to 191,073 cases in 2015 and to 526,841 in September 2022 [27]. The number of people whose HIV status progressed to the AIDS level also increased markedly from 33,491 cases in 2010 to 83,241 in 2015 and 142,009 in September 2022 and of whom 61,192 people have died from AIDS [27]. Of the total number of PLHIV in the country and those who were newly diagnosed every year, the majority are men [28]. For example, in 2022 alone (January-September), 71% of 36,665 new cases were diagnosed in men, which increased from 67% in 2020 and 70% in 2021 [28-30]. Risky sexual behaviours or sexual contacts have been reported as the main mode of transmission, and this is supported by the high prevalence of the infection (85.3%) being diagnosed in the sexually active age group (20-49 years) [28]. The increased HIV infections and AIDS cases in the country seem to reflect the low percentage of ART uptake among PLHIV. Of the total number of PLHIV in the country, 79% (417,778) knew their HIV status and of whom only 41% (169,767) were on ART [28]. Of the ones who were on ART, only 29% (27,381) had their viral load suppressed [28]. The low prevalence of ART uptake seems to also reflect a low coverage of HIV care and treatment services, including HIV testing and ART in Indonesia [28]. It has been reported that HIV care and treatment services are not available in several districts. Of 504 districts in the country, only 476 have the services and provide regular reports on service use [28].

Belu is located in Eastern Indonesia and shares the border with East Timor [31]. It has 12 sub-districts and 81 villages and covers an area of 1,284,94 km2, primarily rural and occupied by a total population of 204,541 people [31]. It has 17 public health centres or sub-public health centres, three private hospitals and one public hospital where the only HIV clinic in the district is located [32]. HIV care services available at the clinic are limited to HIV counselling, testing and antiretroviral therapy (ART). Liver and kidney function tests, CD4 tests and viral load tests to support ART or measure the effectiveness of ART are not available. In terms of HIV infections, it is reported that there had been 1200 cases diagnosed in the district and of whom only about 25% ever started ART at the clinic [31]. The limited availability of HIV-related healthcare services is one of the barriers to HIV testing and ART uptake among general population and PLHIV in the district [3]. Yogyakarta municipality covers urban areas of 46 km2  and is part of the Special Region of Yogyakarta province, Indonesia[33]. It has a total population of 636,660 people [33]. It has a total of 1,488 PLHIV and of whom the majority were on ART [34]. The healthcare facilities available in the district include 20 hospitals and 27 public health centres and sub-public health centres [33]. HIV care services, such as HIV counselling and testing (VCT), CD4 and viral load tests, ART, and other medical tests to support HIV treatment, are provided in ten public health centres and four hospitals [35, 36].

  • We have added the following to the limitation section

In addition, as the focus of the study was on heterosexual men, we did not include high-risk groups, such as men who have sex with men, male sex workers and male drug users, who have been reported to suffer double burden due to the infection and their sexual orientation and activities [60]. They may face different experiences and challenges and have different coping strategies, which are worth exploring in future research.

Comments (on the Quality of English Language)

The paper is reasonably well written but needs further editing. For example, line 49 uses the words "contend the disruption" which should read "contend with the disruption". Line 53 uses the words "presented to their life" and should say "presented to their lives". These errors, especially missing prepositions, and the use of plural pronouns with singular nouns, occur through out the paper. The authors have done a great deal of work without funding for their research, but maybe they can find someone to correct the English grammar and usage.

Response:

  • Thank you very much, we have edited the manuscript entirely.

Reviewer 2 Report

Dear Authors,

It does not happen often that I come across a manuscript as fine as yours. The topic is outstanding and will make significant advances to the literature. Most HIV research is deficit oriented and your new approach opens a whole new window in the lives of men living with HIV. I can see many strength-based interventions in the future.

The description of the background, methods, and findings, as well as an in-depth discussion, was a delight to read and I look forward when this paper is published. I have not found any issues with the paper that I would like to change.

Author Response

Comments and Suggestions for Authors

It does not happen often that I come across a manuscript as fine as yours. The topic is outstanding and will make significant advances to the literature. Most HIV research is deficit oriented and your new approach opens a whole new window in the lives of men living with HIV. I can see many strength-based interventions in the future.

The description of the background, methods, and findings, as well as an in-depth discussion, was a delight to read and I look forward when this paper is published. I have not found any issues with the paper that I would like to change.

Response:

  • Thank you very much for your appreciation

Reviewer 3 Report

This paper examines many interesting and innovative topics, but it’s intent, focus, and conceptual constructs need tightening or better explanation.

·            A worrisome problem lies in the intent behind the paper’s analysis. As the authors no doubt agree, the role of science is without bias or a pre-agenda to investigate what exists.  In contrast, the paper notes that its analysis sets out with the goal to “recast data into a slightly more positive light” than how it currently is conceptualized in the literature. Had  this alternative view emerged from the qualitative data analysis, that’s a new and valid outcome.  But to argue that something is true prior to data analysis as this phrase seems to imply introduces a level of doubt as to the validity of the paper’s findings.

·            Certainly not all the men’s response to coping with the disruptions of an HIV diagnosis were positive, nor is it likely that all the men attempted to or successfully redefined their personal biographies. The validity and rigor of the analysis would be better demonstrated by attention to also relating such alternative actions and outcomes.

·            Despite a demographic table indicating differences within the sample, the analysis treats the study’s sample as if the men are homogeneous in age, demographic backgrounds, and HIV experience.  Yet some unspecified number of sample members likely are men who have sex with me, use drugs, or sell sex (an activity not confined solely to women).  Biographical reinvention likely differs in some fundamental ways based on membership in a HIV key population.  Meanwhile, men at age 20 likely confront different challenges to biographical reinvention than men who are at mid-life or in their 60s.  Do life course norms, expectations, and values enter into and help to shape the reinvention process?  It also seems logical that the number of years since diagnosis partly influence  the timing and other coping factors involved in the self-reinvention process. Are there data available that would address these many biographical differences within the sample?

·            The title of the paper and literature review indicate that the analysis focuses on explaining biographical reinvention based on using  additive versus subtractive strategies, but both the reinvention process and its accompanying strategies are largely buried and not readily apparent within the text’s larger discussion of the role of hope, optimism, and resilience in facilitating this process.  Part of the problem is that a plethora of concepts and assertions compete for major attention throughout the paper and include: “biographical disruption, ” “biographical reinvention,”  “additive and subtractive strategies, “hope” as envisioned by Snyder & colleagues, “optimism” as conceptualized by Seligman including  “learned optimism (although is not clear how it is learned,” “coping,” “motivation to profoundly rethink ideas about the future,” “internal versus external  assets,” and “becoming a more organized person.”  A more focused and possibly more limited use of  explanatory variables might help.  Also the authors might consider adding a conceptual figure/diagram to the paper that visually shows the study's analytic models and how they and all their influential constructs fit together to facilitate or produce “biographical reinvention” as an outcome.

·            The central argument that the paper attempts to advance seems conceptually circular. It appears from the text that in addition to external assets, internal assets (hope, optimism, resilience) allow the men to take on new roles/attitudes (engage in additive strategies) and to give up dysfunctional behavior (engage in subtraction strategies). These additive and subtraction strategies, in turn, allow the men “not only to cope with the HIV repercussions and improve their physical and mental health conditions, but to think or work on a ‘reinvented’ biography which "encompasses resilience, hope and optimism for better health, life and future.”   It seems then, that employing hope, optimism, and resilience leads to a reinvented biography  encompassing resilience, hope, and optimism (the same variables on both sides of the equation). 

·            Reinvention is a process, yet discussion and examples in the text don’t directly address the steps and phases of transformation that constitute the pathway from one biographical stage to another.  How do the men see themselves initially? What does their self-assessment become after reinvention takes place? 

·            None of the 3 concepts (hope, resilience, optimism) appear to be  assets consciously employed by the men to adopt different behavior/activities or to redefine their concept of self or presentation of self  to others.  Rather they seem to be emotions that motivate and help them to make life changes in coping with HIV.  Please clarify if the process of “biographical reinvention” is  something the men consciously do or if it is merely a latent consequence of coping with the challenges and  demands of their illness.

·            The study’s thematic analysis was informed by the Ritchi and Spencer frameworks (both of which deal with emotions) plus its  core focus builds on  Bury’s concept of “biographical disruption” ( a process),  and  Ward’s concept of “additive” and “subtractive strategies” (behavioral action).  Although it  attempts to do so, the paper needs to more convincingly argue how these 4 conceptual approaches intersect with and complement each other.  Currently, each approach seems somewhat grafted on to the others without coalescing into a compelling overall approach. Note also that the conclusion section refers to the value of using an “asset-based approach” to analyze the data, and not the full range of the 4 approaches/models that supposedly drove the analysis.

·            Despite its title and text maintaining that the analysis focuses on biographical reinvention, the paper really seems to be about coping with the disruptions precipitated by HIV (see for example sentences 22-23 in the abstract as one example of the many sentences that appear to mention coping as driving the change that follows diagnosis). From the perspective of coping, other variables in the current analysis seem better argued as facilitators (assets etc.) or hindrances (not discussed in the current version) as part of the coping process.  Biographical reinvention seems a possible  latent consequence of such coping rather than the men’s intended or conscious goal. If so, the title and primary focus of the paper, including its literature review, should be revised to argue this premise. If not, the current argument needs to be made more convincingly.

·            Section 2.2 that describes the study’s two recruitment settings includes a lot of information about the 2 clinic sites, only some of which pertains to HIV or the men’s process of reinvention. This section could be condensed by removing such unnecessary information.

·            “Snowball sampling” refers to recruitment methods in which study participants aid in recruiting additional informants. Instead of utilizing SS, “convenience sample” is probably the more accurate term for recruitment in this study.

·            The number of references cited seems overdone. Numbers 26-30 relate to properties of the recruitment sites yet, as mentioned above, not all the information that they convey seems necessary or relevant to this study.

·            The text is well written. but take “rheumatoid arthritis” out of sentence 44 – it doesn’t add anything and is distracting.

Round 2

Reviewer 3 Report

The revised paper is much improved.  Positive changes include:

Some grammatical editing is needed.

Author Response

Response to reviewers' file is attached.
